# Human disturbance increases spatiotemporal associations among mountain forest terrestrial mammal species

Xueyou Li[1]*, William V Bleisch[2], Wenqiang Hu[1], Quan Li[1], Hongjiao Wang[1], Zhongzheng Chen[3], Ru Bai[1], Xue-Long Jiang[1]*

[1]State Key Laboratory of Genetic Resources and Evolution & Yunnan Key Laboratory of Biodiversity and Ecological Conservation of Gaoligong Mountain, Kunming Institute of Zoology, Chinese Academy of Sciences, Kunming, China; [2]China Exploration and Research Society, 2707-08 SouthMark, Wong Chuk Hang, Hong Kong, China; [3]Anhui Provincial Key Laboratory of the Conservation and Exploitation of Biological Resources, College of Life Sciences, Anhui Normal University, Wuhu, China

**\*For correspondence:**
lixueyou@mail.kiz.ac.cn (XL);
jiangxl@mail.kiz.ac.cn (X-LongJ)

**Competing interest:** The authors declare that no competing interests exist.

**Abstract** Spatial and temporal associations between sympatric species underpin biotic interactions, structure ecological assemblages, and sustain ecosystem functioning and stability. However, the resilience of interspecific spatiotemporal associations to human activity remains poorly understood, particularly in mountain forests where anthropogenic impacts are often pervasive. Here, we applied context-dependent Joint Species Distribution Models to a systematic camera-trap survey dataset from a global biodiversity hotspot in eastern Himalayas to understand how prominent human activities in mountain forests influence species associations within terrestrial mammal communities. We obtained 10,388 independent detections of 17 focal species (12 carnivores and five ungulates) from 322 stations over 43,163 camera days of effort. We identified a higher incidence of positive associations in habitats with higher levels of human modification (87%) and human presence (83%) compared to those located in habitats with lower human modification (64%) and human presence (65%) levels. We also detected a significant reduction of pairwise encounter time at increasing levels of human disturbance, corresponding to more frequent encounters between pairs of species. Our findings indicate that human activities can push mammals together into more frequent encounters and associations, which likely influences the coexistence and persistence of wildlife, with potential far-ranging ecological consequences.

## eLife assessment

In this study, camera trapping and species distribution models are used to show that human disturbance in mountain forests in the eastern Himalayas pushes medium-sized and large mammal species into narrower habitat space, thus increasing their co-occurrence. While the collected data provide a **useful** basis for further work, the study presents **incomplete** evidence to support the claim that increased co-occurrence may indicate positive interactions between species.

## Introduction

Pervasive human activities can disrupt invisible facets of biodiversity such as species associations, with potential cascading ecosystem effects (*Naeem et al., 1994*; *Parsons et al., 2022*). Human encroachment into natural ecosystems squeezes the spatiotemporal niches of wildlife species (*Gilbert et al.,*

*2022*), altering the number and magnitude of associations in a community (*Burkle et al., 2013*), and accelerating species decline and loss (*Jones et al., 2018*). Humans play a role as 'super predators' in shaping the co-occurrence of other species with complicated indirect modifications to multiple interactions between organisms (*Gilbert et al., 2022*; *Moll et al., 2021*). Human disturbance, such as chronic landscape modification and acute direct human presence, may significantly alter spatio-temporal distribution of species and fundamentally change the way that species interact (*Li et al., 2022a*; *Sévêque et al., 2020*). For example, expanding human footprints have compressed the space and time available for mammals to share by restricting animal movements (*Tucker et al., 2018*) and increasing wildlife nocturnality (*Gaynor et al., 2018*). As human and animal activities increasingly overlap in time and space, it is important to assess and quantify the potential for human-induced changes in species association to ecosystem structure and function (*Penjor et al., 2022*).

Species are not distributed independently of each other; rather, they co-occur in time and space and interact (*Tilman, 1994*; *Wisz et al., 2013*). The spatiotemporal co-occurrence of species, termed interspecific associations, provides unique ecological information and has important consequences for ecosystem integrity (*Gorczynski et al., 2022*; *Keil et al., 2021*). However, segregated species co-occurrence could be generated by processes such as negative interspecific interaction, distinct environment requirements, and dispersal limitations; similarly, aggregated species pairs may reflect positive interspecific interaction but could also reflect shared environmental preferences (*Kathleen Lyons et al., 2016*; *Song et al., 2020*). Although interspecific co-occurrence or avoidance cannot be used to directly estimate species interactions (*Blanchet et al., 2020*), animals cannot interact if their spatiotemporal niches do not overlap (*Gilbert et al., 2022*) and strong interactions should be expected to lead to significant associations. Thus, interspecific associations convey key information about interactions between sympatric species (*Boron et al., 2023*).

Disruptions to the spatiotemporal relationships of species can result in serious ecological consequences including alteration of community structure (*Tulloch et al., 2018*), upsetting the competitive balance between species (*Boron et al., 2023*), increasing disease transmission (*Hassell et al., 2017*), and accelerating local extinction (*Fidino et al., 2019*). This can significantly distort the distribution of ecological functions that species provide, ultimately influencing ecosystem dynamics (*Gardner et al., 2019*). Taken as a whole, interactions between sympatric species play a fundamental role in community assembly and are intricately related to ecosystem stability and resilience (*Boron et al., 2023*). Thus, understanding how interspecific spatiotemporal associations change across human disturbance gradients provides valuable insight into the long-term implications of human impacts on ecosystem function and recovery relevant to biodiversity conservation in the Anthropocene.

Species interactions are known to be context-dependent such that they can vary across space and time, for example along environmental gradients (*Chamberlain et al., 2014*; *Davis et al., 2018*; *Pellissier et al., 2018*; *Perrin et al., 2022*). For example, gradients in stress are associated with variation in the outcomes of pairwise species interactions (*Chamberlain et al., 2014*). A key challenge in community ecology is to identify the conditions under which negative and positive species interactions are more likely to occur. For example, the controversial stress-gradient hypothesis predicts that positive interactions should increase as environmental conditions become more severe. Alpine bird communities have been shown to have a higher frequency of positive associations in grasslands with low productivity compared with forests with high productivity (*García-Navas et al., 2021*) and Savanah ungulates are more likely to form mixed-species groups in areas where there is predation risk (*Beaudrot et al., 2020*). Increasingly, major stresses on wild communities derive from human activity, and understanding how species interactions vary in response to human disturbance is pivotal in making robust ecological predictions about biodiversity responses to changing environmental conditions (*Dangles et al., 2018*; *Perrin et al., 2022*). However, a review of the literature indicates that the impact of human activity on interspecific interactions of terrestrial mammals remains poorly understood (*Boron et al., 2023*), particularly in mountain forests where anthropogenic impacts are often pervasive and increasing.

Medium and large-sized terrestrial mammals are key components of mountain forest communities. They play crucial roles in maintaining biodiversity and ecosystem functions (*Lacher et al., 2019*), and are especially vulnerable to anthropogenic activities (*Li et al., 2022b*). Here, we set out to investigate the spatial and temporal patterns of occurrence and the interspecific associations within a terrestrial mammalian community along human disturbance gradients. We executed a systematic camera

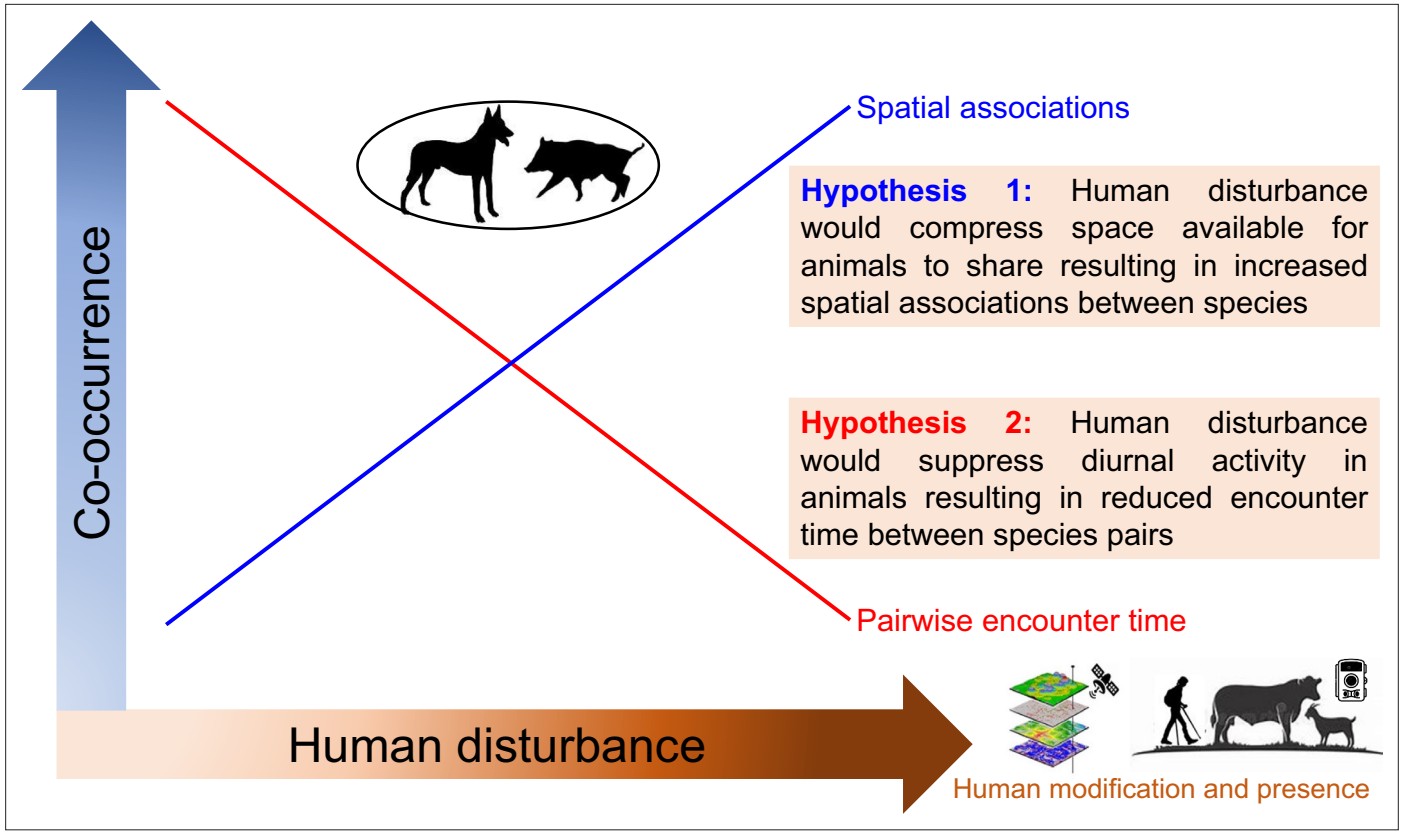

**Figure 1.** Conceptual framework illustrating the community-level effects of human disturbance on spatiotemporal associations among mountain forest terrestrial mammal species.

trapping survey spanning 4100 km$^2$ of the Yarlung Zangbo Grand Canyon National Nature Reserve in the eastern Himalayas at a total of 322 camera trapping stations, obtaining detections of both wildlife and humans. We classified human disturbances into two groups: human modification (i.e. relating to habitat modifications by humans) or human presence (i.e. referring to the direct presence of humans) disturbances. We employed a context-dependent joint species distribution model (JSDM; *Tikhonov et al., 2017*) to infer pairwise species associations along human disturbance gradients. We used kernel density distributions of animal diel activity and time between detections of species pairs (time-to-encounter) to compare temporal associations at low and high human disturbances. We consider two hypotheses regarding temporal and spatial interspecific associations. The spatial compress hypothesis posits that human modification of habitats would suppress space available for animals to share (*Tilman et al., 2017*), resulting in increased positive spatial associations between species. The temporal compress hypothesis postulates that fear of direct human presence would compress diurnal activity in animals (*Gaynor et al., 2018*), resulting in reduced encounter time (or increased encounter rate) between species (*Figure 1*). Our study incorporates two different types of human disturbances in the analysis to elucidate the effects of humans on multidimensional (i.e. space and time) species associations. We then consider implications for conservation.

## Results

Our camera traps obtained 10,388 independent detections of 17 focal species (12 carnivores and five ungulates) from 322 stations over 43,163 camera days of effort (*Table 1*). We documented a number of species and subspecies of conservation concern, including Bengal tiger *Panthera trigris*, clouded leopard *Neofelis nebulosa,* and dhole *Cuon alpinus*. We also captured 2224 independent detections of humans during the survey period.

**Table 1.** Independent detection of ground-dwelling medium- and large-bodied mammal species based on camera trapping survey in the Yarlung Zangbo Grand Canyon, southeast Tibet.

| Order | Family | Genus | Species | Independent detections | IUCN category |
|---|---|---|---|---|---|
| Cetartiodactyla | Bovidae | *Budorcas* | *Budorcas taxicolor* | 92 | VU |
| Cetartiodactyla | Bovidae | *Capricornis* | *Capricornis milneedwardsii* | 2992 | NT |
| Carnivora | Felidae | *Catopuma* | *Catopuma temminckii* | 232 | NT |
| Carnivora | Canidae | *Cuon* | *Cuon alpinus* | 256 | EN |
| Carnivora | Mustelidae | *Martes* | *Martes flavigula* | 469 | LC |
| Cetartiodactyla | Cervidae | *Muntiacus* | *Muntiacus muntjak* | 4696 | LC |
| Cetartiodactyla | Bovidae | *Naemorhedus* | *Naemorhedus baileyi* | 254 | VU |
| Carnivora | Felidae | *Neofelis* | *Neofelis nebulosa* | 45 | VU |
| Carnivora | Viverridae | *Paguma* | *Paguma larvata* | 223 | LC |
| Carnivora | Felidae | *Panthera* | *Panthera tigris* | 26 | EN |
| Carnivora | Felidae | *Pardofelis* | *Pardofelis marmorata* | 54 | NT |
| Carnivora | Felidae | *Prionailurus* | *Prionailurus bengalensis* | 164 | LC |
| Carnivora | Prionodontidae | *Prionodon* | *Prionodon pardicolor* | 28 | LC |
| Cetartiodactyla | Suidae | *Sus* | *Sus scrofa* | 269 | LC |
| Carnivora | Ursidae | *Ursus* | *Ursus thibetanus* | 463 | VU |
| Carnivora | Viverridae | *Viverra* | *Viverra zibetha* | 37 | LC |
| Carnivora | Canidae | *Vulpes* | *Vulpes vulpes* | 88 | LC |

## Species-specific response to habitat covariates

Species varied in their responses to habitat covariates. Nine out of 17 species showed a strong positive response to forest cover (*Figure 2*, *Figure 2—source data 1*). Human presence had apparent negative effects on the occurrence of clouded leopard (mean = –0.967, 95% CI=−2.35 to −0.025), taking *Budorcas taxicolor* (mean = –0.449, 95% CI=−0.935 to −0.071) and red goral *Naemorhedus baileyi* (mean = –0.806, 95% CI=−0.935 to −0.071), but was apparently positively associated with the occurrence of wild boar *Sus scrofa* (mean = 0.521, 95% CI=0.045 to 1.093) and golden cat *Catopuma temminckii* (mean = 0.427, 95% CI=0.165 to 0.804; *Figure 2—source data 1*). Human modification was apparently negatively associated with the occurrence of dhole (mean = –0.24, 95% CI=−0.493 to -0.027), red fox *Vulpes vulpes* (mean = –1.588, 95% CI=−2.656 to −0.769), Asiatic black bear *Ursus thibetanus* (mean = –0.234, 95% CI=−0.42 to −0.056), red goral (mean = –0.665, 95% CI=−1.129 to −0.238) and Mainland serow *Capricornis sumatraensis* (mean = –0.477, 95% CI=−0.697 to −0.26), but was positively apparently associated with muntjac *Muntjac* spp. (mean = 3.632, 95% CI=0.946 to 9.999), masked palm civet *Paguma larvata* (mean = 2.076, 95% CI=1.26 to 3.145), large Indian civet *Viverra zibetha* (mean = 0.782, 95% CI=0.175 to 1.558) and marbled cat *Pardofelis marmorata* (mean = 0.689, 95% CI=0.232 to 1.236; *Figure 2*, *Figure 2—source data 1*).

## Effects of human disturbances on spatial co-occurrence

Out of the 136 estimated pairwise residual correlation coefficients in occupancy, 87 (64 %) were positive at lower human modifications (*Figure 3a*). At the moderate and higher modifications, the species pairs with positive associations increased to 107 (79 %) and 118 (87%), respectively (*Figure 3b and*

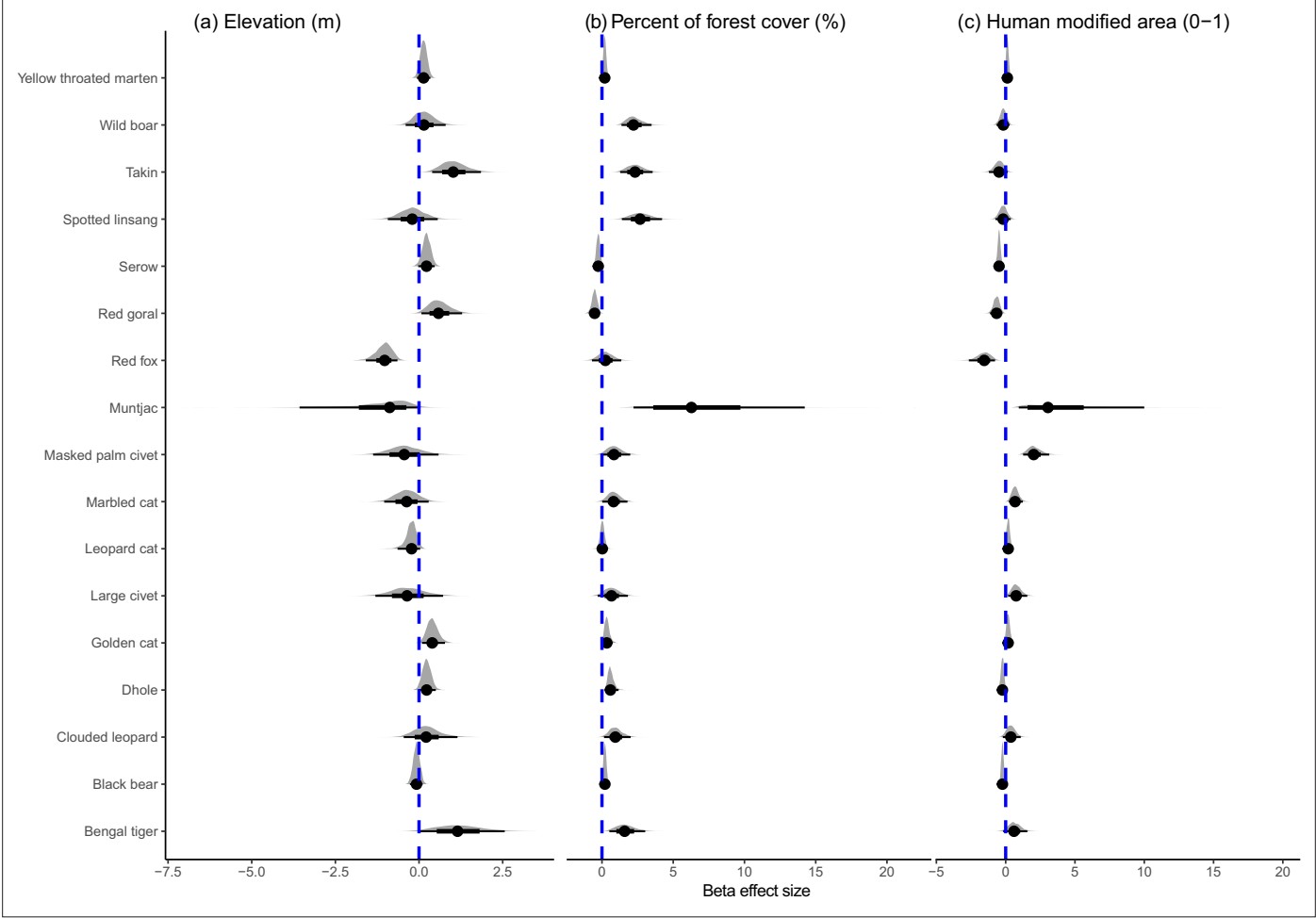

**Figure 2.** The effects of environmental and anthropogenic variables on terrestrial mammals in the Yarlung Zangbo Grand Canyon.

The online version of this article includes the following source data for figure 2:

**Source data 1.** Standardized beta coefficients, and 95% credible intervals, for the influence of anthropogenic and environmental covariates on the probability a species used an area during our camera-trap survey in Medog region.

*c*). At lower modifications, correlation coefficients for 18 species pairs were positive and had a 95 % CI that did not overlap zero (*Figure 3—figure supplement 1*), and the number increased to 65 in moderate modifications (*Figure 3—figure supplement 2*) but dropped to 29 at higher modifications (*Figure 3—figure supplement 3*).

Along human presence gradients, 88 pairwise residual correlation coefficients (65%) at lower human presence habitats were positive or close to neutral (*Figure 4a*). At the moderate and higher human presence habitats, the species pairs with positive associations increased to 115 (85%) and 113 (83%), respectively (*Figure 4b and c*). The significant positive associations at low, moderate, and higher human presence habitats were 6 (4%, *Figure 4—figure supplement 1*), 76 (56%, *Figure 4—figure supplement 2*), and 44 (32%, *Figure 4—figure supplement 3*), respectively.

## Effects of human disturbances on temporal co-occurrence

Human presence was associated with significantly increased nocturnality of carnivores (mean = 0.163, 95% CI=0.089 to 0.236, *Figure 5a*), but showed no significant effects on ungulates (mean = –0.004, 95% CI=–0.031 to 0.023, *Figure 5b*) and combination of guilds (mean = –0.001, 95% CI=–0.028 to 0.026; *Figure 5c*). Human modification had no strong effects on nocturnality of any guilds and combination of guilds (i.e. 95% CIs include zero, *Figure 5—figure supplement 1*).

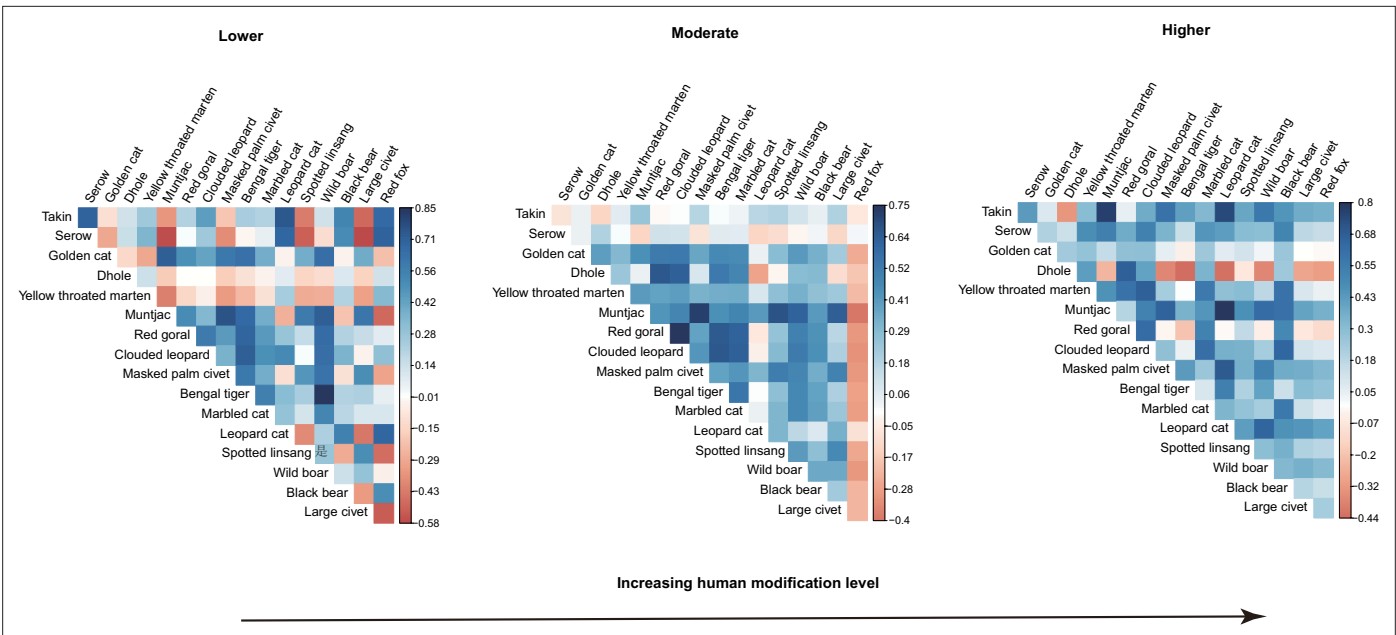

**Figure 3.** Estimates of associations between 17 terrestrial mammals across camera trapping stations with different human modifications in the Yarlung Zangbo Grand Canyon. Associations are shown for the region's (**a**) minimum (Lower), (**b**) mean (Moderate), and (**c**) maximum (Higher) human modifications.

The online version of this article includes the following figure supplement(s) for figure 3:

**Figure supplement 1.** 95% confidence intervals of residual associations between species pairs at lower human modification level.

**Figure supplement 2.** 95% confidence intervals of residual associations between species pairs at moderate modification level.

**Figure supplement 3.** 95% confidence intervals of residual associations between species pairs at higher modification level.

Both human modification and human presence were associated with significantly reduced time between detections of pairs (human modification: mean = −1.07, 95% CI=−1.39 to −0.73; human presence: mean = −0.69, 95% CI=−0.92 to −0.46; *Figure 6*).

## Discussion

Despite accumulating evidence of widespread impacts of humans on wildlife distribution and activity patterns, our understanding of how different types of anthropogenic pressures reshape species associations remains limited (*VanScoyoc et al., 2023*). Here, we compared the impacts of human modification (spatial compress effects) and human presence (temporal compress effects) on spatio-temporal associations among threatened terrestrial mammals based on systematic camera-trapping data in an understudied Himalayan landscape. Our results show a strong influence of humans on species co-occurrence patterns. The overall results of the spatiotemporal associations across levels of human modification and human presence suggest that humans are associated with increasing positive spatiotemporal associations among species. Specifically, we detected a higher incidence of positive associations in habitats with moderate and higher levels of human modification and human presence compared to those located in habitats with lower disturbance levels (*Figures 3 and 4*). On the temporal axis, we detected a significant reduction of pairwise encounter time at increasing levels of human disturbance (*Figure 5*). Our results, therefore, demonstrate that human disturbance can upset interspecific associations both on spatial and temporal niche dimensions.

Interspecific spatial associations can arise from species interaction, response to environmental covariates, and common dispersal barriers (*Blanchet et al., 2020*; *de Jonge et al., 2021*; *Poggiato et al., 2021*). Our context-dependent JSDM, which accounts for similarities and disparities in species-specific response to habitat covariates considered relevant to medium- and large-bodied mammal species, detected a higher prevalence of interspecific spatial associations in human-disturbed habitats. Humans can have 'bottom-up' impacts on animal distribution and associations by land use

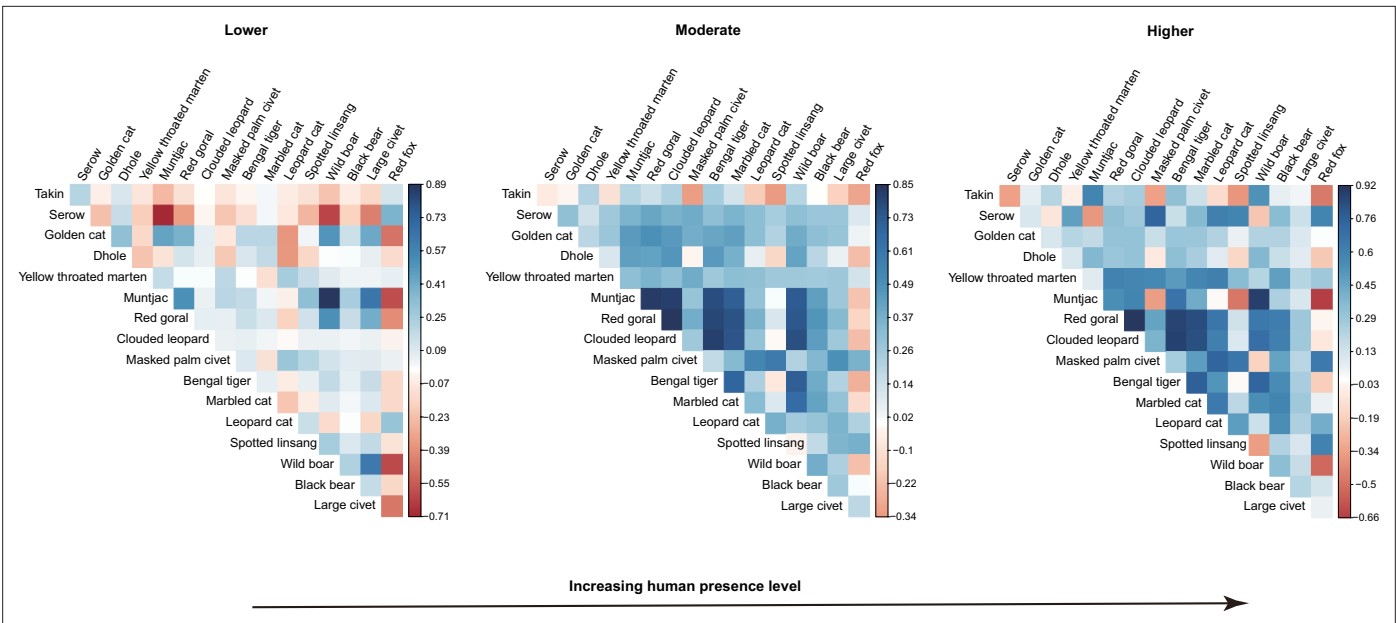

**Figure 4.** Estimates of associations between 17 terrestrial mammals across camera trapping stations with different human presence in the Yarlung Zangbo Grand Canyon. Associations are shown for the region's (**a**) minimum (Lower), (**b**) mean (Moderate), and (**c**) maximum (Higher) human presence.

The online version of this article includes the following figure supplement(s) for figure 4:

**Figure supplement 1.** 95% confidence intervals of residual associations between species pairs at lower human presence level.

**Figure supplement 2.** 95% confidence intervals of residual associations between species pairs at moderate human presence level.

**Figure supplement 3.** 95% confidence intervals of residual associations between species pairs at higher presence level.

change and habitat modification (***Riggio et al., 2020***; ***Tucker et al., 2018***). Human activities can also have top-down impacts on spatiotemporal associations among animals by directly or indirectly changing the landscape of fear (***Palmer et al., 2022***; ***Suraci et al., 2019***). Previous studies suggest that human disturbance may compress the space and time available for communities to use, resulting in an increased frequency of positive associations (***Gilbert et al., 2022***; ***Sévêque et al., 2022***). We interpret our results to mean that anthropogenic presence and disturbance reduce available habitats for wildlife, causing a greater predominance of positive associations in anthropogenic landscapes. On the one hand, human presence, such as while grazing livestock and gathering resources, generates landscapes of fear for wildlife (***Gaynor et al., 2019***; ***Palmer et al., 2022***; ***Suraci et al., 2019***) and may downgrade habitat quality by overexploitation (***Filazzola et al., 2020***). On the other hand, human modification such as land-use change often constrains the realized niche space of wildlife, restricting animal movement (***Smith et al., 2018***; ***Tucker et al., 2018***). Overall, our study provides one of the first tests of whether positive spatiotemporal associations between terrestrial mammals increase along gradients of different types of anthropogenic pressures. Our results add to the growing body of evidence that suggests anthropogenic activities reduce available niche space for animals, causing observed positive spatial associations among species (***Gilbert et al., 2022***; ***Gorczynski et al., 2022***; ***Murphy et al., 2021***).

Identifying thresholds of anthropogenic activity that shift species behavior and co-occurrence will be key to drawing useful inferences from human impact studies and improving our knowledge on when altered associations may lead to reverberating impacts on ecosystems (***VanScoyoc et al., 2023***; ***Wilson et al., 2020***). We note that the number of species pairs with significant positive associations rapidly increased from lower to moderate levels of human disturbance, but dropped from mean to higher levels of human disturbance (***Figure 3—figure supplements 1–3*** and ***Figure 4—figure supplements 1–3***). Such patterns are consistent across human modification and human presence levels, indicating a threshold after which the positive effects of human disturbance on species associations were dampened. Human disturbance is an important factor in shaping species distributions (***Jones et al., 2018***; ***Samia et al., 2015***). Wildlife often has a limited tolerance threshold to human

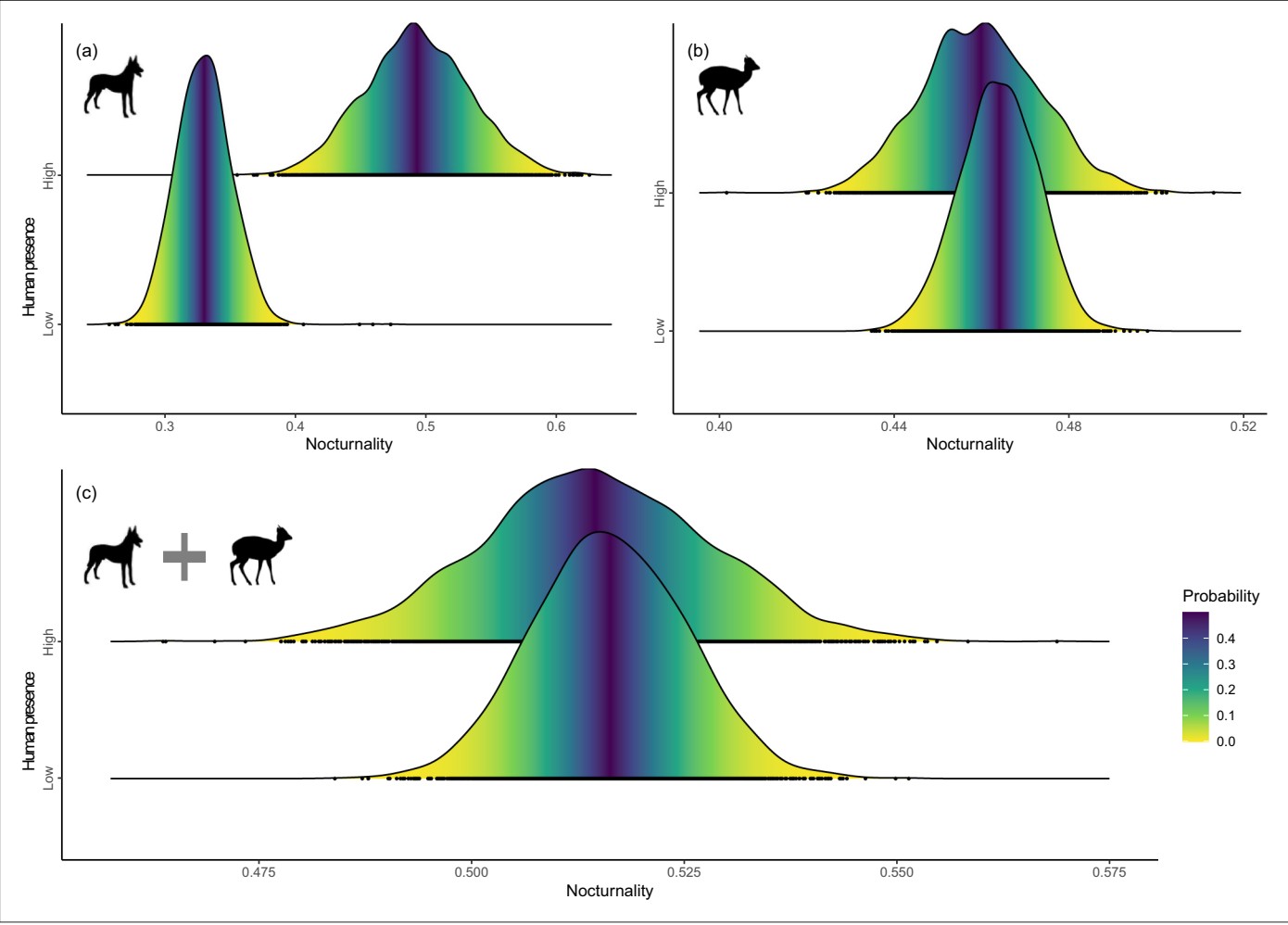

**Figure 5.** Density distributions of encounter time between successive detections of species pairs (in log-transformed days) in low- and high-human modification (**a**) and human presence (**b**) habitats, and differences in time-to-encounter between species pairs in low- and high-human modification (**c**) and human presence (**d**) habitats. The solid vertical lines in (**c**) and (**d**) represent mean differences, and the dashed vertical lines indicate 95% confidence intervals.

The online version of this article includes the following figure supplement(s) for figure 5:

**Figure supplement 1.** Shifts in nocturnality of carnivores (**a**), ungulates (**b**), and combination of carnivores and ungulates (**c**) in the lower- and higher-human modification habitats.

activity (**Polaina et al., 2018**; **Samia et al., 2015**; **Smith et al., 2019**). When human pressures reach levels that preclude a species from occurring at a site (**Polaina et al., 2018**), prediction of species associations in an anthropogenic context may become uninformative. More work needs to be done to further clarify the mechanisms driving the observed co-occurrence patterns.

Anthropogenic activities can also shift temporal niche of animals (**Gaynor et al., 2018**; **Li et al., 2022a**), possibly altering encounter rates among species and trophic dynamics that structure communities (**Gilbert et al., 2022**; **Karanth et al., 2017**; **Mills and Harris, 2020**). Congruent with our hypothesis, we observed a significant reduction of pairwise encounter time at increasing levels of human disturbance, corresponding to more frequent encounters between the pairs (**Gilbert et al., 2022**). Although we did not find a significant shift in nocturnality at community level, we detected a significant shift to nocturnal activity of carnivores, indicating different sensitivities to human presence among the carnivores and ungulates in our study system. Fear of humans may explain the temporal response of wildlife to human presence. Numerous studies have shown that fear of humans as 'super predators' can have suppressive effects on wildlife activity, which may contribute to increased temporal overlap among species (e.g. **Sévêque et al., 2022**; **Suraci et al., 2019**).

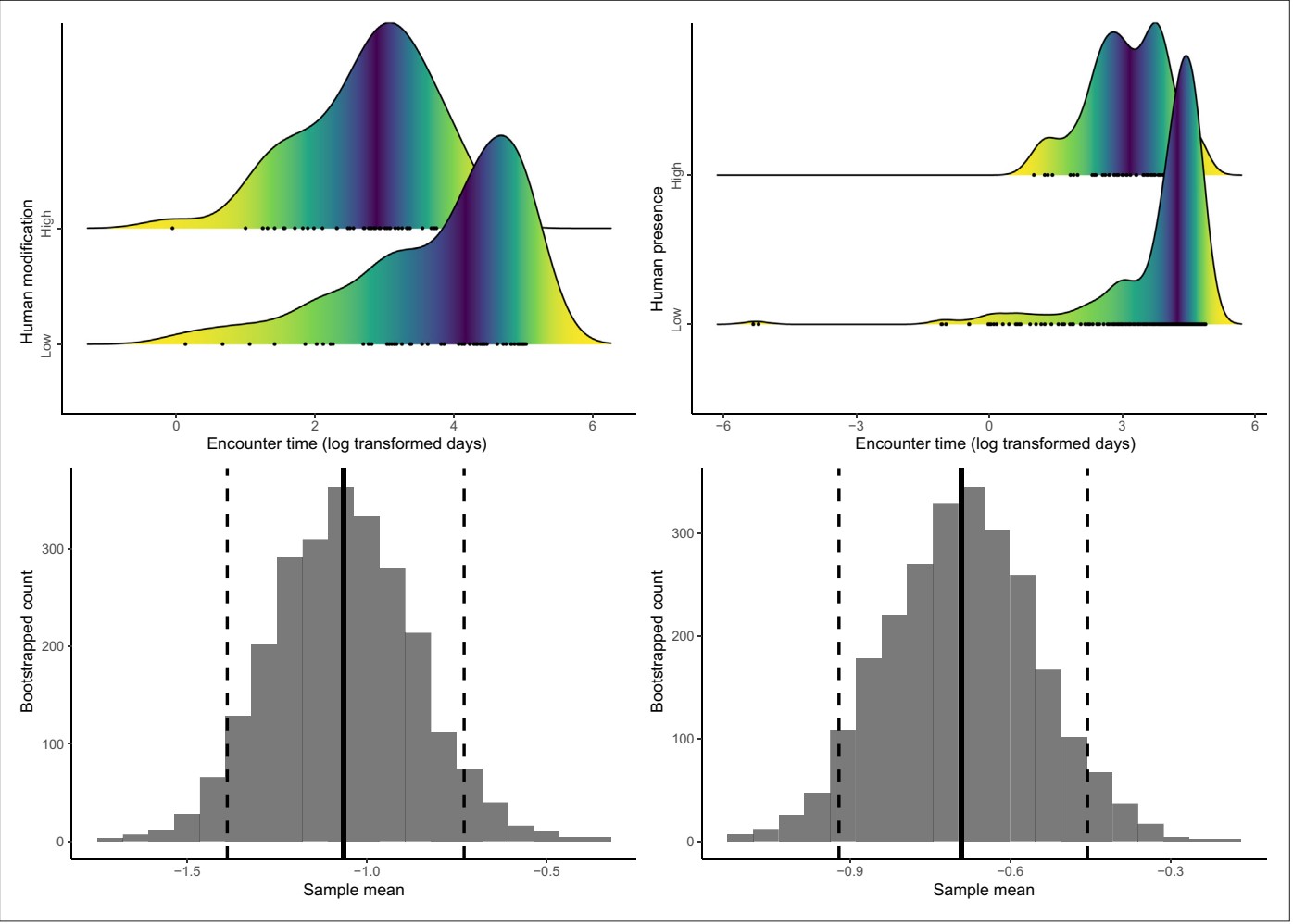

**Figure 6.** Location of study area in the Yarlung Zangbo Grand Canyon National Nature Reserve in the southeast of the Tibetan Autonomous Region of China.

All in all, our results indicate that both human modification and human presence may rewire species interactions by increasing spatial and temporal co-occurrence. Studies on the impacts of humans on wildlife communities should explicitly account for different types of co-occurring disturbances (*Li et al., 2022a*). Our camera trapping survey observed that human presence is pervasive even inside this remote protected area in the Tibetan Autonomous Region, indicating that wildlife and their habitats are exposed to frequent human disturbance. Humans exploit resources in protected areas in many ways, including through livestock herding, resource gathering, illegal hunting, and recreation, all of which impact wildlife and their habitats to varying degrees (*Harris et al., 2019*; *Mills and Harris, 2020*). Our results demonstrate prevalent disruptions to species co-occurrence patterns from humans. If we are to preserve biodiversity in protected areas, we must work to understand the negative effects on multiple facets of biodiversity from co-occurring anthropogenic pressures. Only then can we design effective mitigation measures.

Although interspecific associations should not be directly interpreted as a signal of biotic interactions between pairs of species (*Blanchet et al., 2020*; *Poggiato et al., 2021*), it describes a unique facet of biodiversity, and can provide important insights into the coexistence and persistence of wildlife as well as ecosystem function in settings with anthropogenic activity (*Keil et al., 2021*; *Lai et al., 2020*; *VanScoyoc et al., 2023*). Since interspecific associations often increases with the ecological similarity of species involved (*Gorczynski et al., 2022*), the tendency of many species towards positive associations with increase in anthropogenic pressures indicates a trend towards homogenization of terrestrial mammal communities. Such human-mediated changes in species co-occurrence patterns

can have serious ecological consequences at multiple scales (*Gilbert et al., 2022*; *Gorczynski et al., 2022*). For individual species, increased positive associations may affect fitness and population dynamics, increasing local extinction rates (*Kuussaari et al., 2009*; *Parsons et al., 2022*). Increase in spatial aggregation among functionally similar species may be indicative of a hidden extinction debt (*Kuussaari et al., 2009*). At the community level, increased positive associations may depress co-occurrence network complexity and stability, amplify interactions such as predation, and simplify communities with similar traits or co-occurrence patterns (*Manlick and Pauli, 2020*; *Mills and Harris, 2020*). Also, increased encounter rates between species may expand disease transmission across communities (*Hassell et al., 2017*). In addition, wildlife can host a variety of zoonotic diseases (*Wicker et al., 2017*) and populations unnaturally associated with humans are more likely to transmit pathogens (*Jones et al., 2013*). We observed that several species such as masked palm civet and wild boar are positively associated with humans in our study area, indicating a substantial overlap between their habitats and human activity. Ongoing habitat modification, livestock grazing, and resource gathering in this region may increase probability of pathogen exchange.

The effects of human disturbance on species associations might be scale-dependent (*Gilbert et al., 2022*). At present, our sampling design only considers the effects of cumulative human modification and instantaneous local human presence on spatial associations and temporal encounter time at each camera trapping station. We are not able to compare the effects of human disturbance at broader scales (e.g. landscape scale) as such an analysis requires adequate data from hierarchical samplings. Estimating the effects of human disturbance at different spatial scale on species associations is a promising approach. Thus, we encourage future work to further clarify the multiscale ecological effects of human disturbance on species associations.

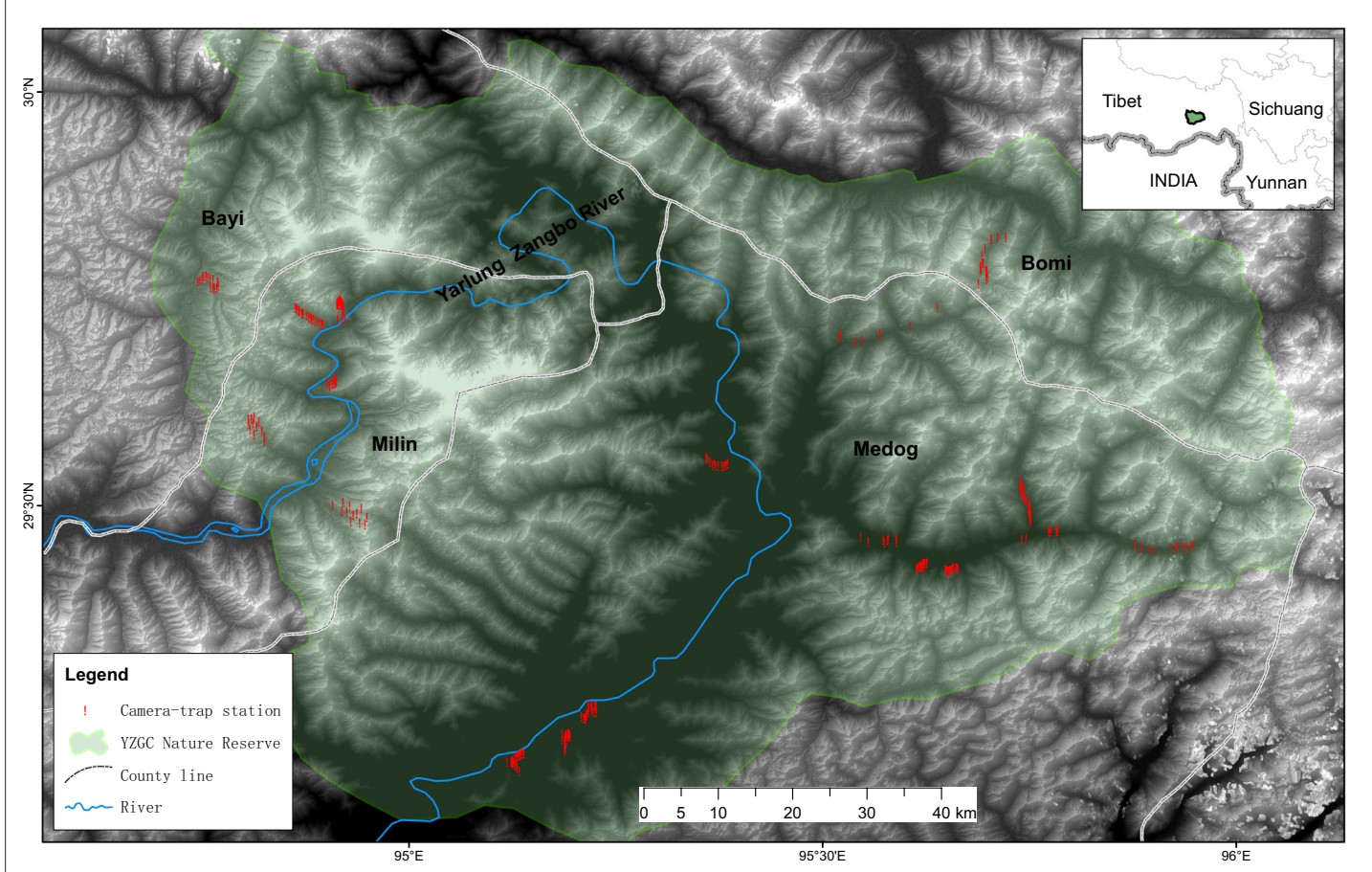

**Figure 7.** Location of study area in the Yarlung Zangbo Grand Canyon National Nature Reserve in the southeast of Tibetan Autonomous Region of China.

In conclusion, our study demonstrated that anthropogenic pressures increase spatiotemporal associations of terrestrial mammals from low to mean levels of human disturbances, but the frequency of positive spatial associations dropped from mean to higher levels of human disturbances. Such variations in species associations likely influence the coexistence and persistence of wildlife, with potentially far-ranging ecological consequences. Because terrestrial mammals like carnivores and ungulates play fundamental roles in regulating montane forest ecosystems, the prevalent disruptions to their associations may precipitate biodiversity loss and impair ecosystem function. With increasing human presence and human modification of areas throughout the world, identifying thresholds of anthropogenic activity that shift species relationships, limiting human activity, and increasing landscape connectivity across protected areas may be imperative to maintain interspecific spatiotemporal associations that underpin ecosystem resilience. Moreover, the methods we applied highlight the utility of camera trapping surveys in studying the spatiotemporal relationships among elusive species in settings with anthropogenic activities.

## Materials and methods

### Study area

The study was carried out inside the Yarlung Zangbo Grand Canyon National Nature Reserve (29°05′–30°02′ N, 94°39′–96°6′ E) in Nyingchi City in the southeast Tibet Autonomous Region of China (*Figure 7*). This area is situated within the Eastern Himalaya Biodiversity Hotspot, a globally important region for biodiversity conservation (*Li et al., 2021*). The Yarlung Zangbo Grand Canyon is the deepest in the world, with an elevation drop of more than 7000 m, and has the reputation of 'Gene Bank of Mountain Biological Resources' and 'Natural Vegetation Museum' (*Duan et al., 2022*). This region is characterized by dramatic vertical zonation of vegetation. From the valley bottom to the mountain peak, the main vegetation types consist of low mountain tropical monsoon rain forest, subtropical mountain evergreen broad-leaved forest, mid-mountain warm coniferous forest, sub-alpine cold coniferous forest, alpine subrigid shrub meadow, and periglacial alpine vegetation (*Deng et al., 2011*). The complete landscapes in the region harbors endangered species such as Bengal tiger *Panthera tigris tigris* (*Li et al., 2023*). Human activities such as decentralized residential settlements and free-ranging livestock grazing are prevalent in the region, even within the protected area (*Li et al., 2021*).

### Camera trap sampling

Camera trap detections of terrestrial mammals were collected during the dry season between November 2020 and April 2021 and November 2021 and April 2022 to avoid the heavy rainy seasons in the region. The mean trapping efforts were 134 days (89–147 days range). We used Yianws L720 camera traps to conduct the survey. To determine how anthropogenic factors shape spatiotemporal associations among terrestrial mammals, we set up camera trapping stations within the nature reserve based on the intensity of human activities and distance to nearest human settlement. We deployed 350 camera stations in the reserve with different degrees of anthropogenic disturbance, maintaining at least 800 m between them (range 886–2233 m, median 1219). This distance may not satisfy the assumption of population closure and there may be some degree of pseudo-replication as observations of wide-ranging animals may not be independent. For these species, the occupancy estimate can be thought of as an estimate of the probability that the species used the area where the camera trapping station was located, rather than true occupancy (*Li et al., 2018*). Our camera trapping stations spanned a gradient of forest areas with varying levels of human activities and habitat modifications in the surrounding area. We affixed camera traps to trees between 80 and 100 cm off the ground, and they were not baited. Camera sensitivity was set to 'low' to reduce false detections triggered by nonanimal movements. We set cameras to take three photos per detection event, with 3 s delay between subsequent detections. Camera trap photos were later identified to species when possible. We combined all human presence photos into a single 'Human' categorization representing a variety of human activities detected around a camera station (e.g. resource gathering, livestock grazing, recreation, etc.). All photos of the same species (including humans) at the same camera station were considered independent detections if separated by at least 1 hr (*Li et al., 2021*). Several camera stations were invalid due to camera malfunctions or lost cameras. The total valid sampling effort was 43,163 camera days from 322 camera stations that operated effectively. The target species consisted

of ground-dwelling mammal species observed in more than 10 camera stations and weighing more than 1 kg. Thus, the occurrence dataset consists of presence-absence information on 17 mammal species at 322 camera stations.

## Anthropogenic and habitat covariates

We derived two different types of human impacts: chronic human modification (e.g. settlement, transportation night-time lights, etc.) and acute direct human presence (occurrence of people and domestic animals detected by camera traps) to address our hypotheses. We explored the degree of habitat modification based on the Human modification (HM) map metric (**Kennedy et al., 2019**). The HM metric provides a cumulative estimate of artificial modification of terrestrial landscapes based on 13 anthropogenic stressors caused by five human activities (human settlement, agriculture, transportation, energy production, and electrical infrastructure) at a resolution of 1 km (**Kennedy et al., 2019**). The metric is based on both the intensity and extent of impact of each anthropogenic stressor and ranges from 0 (no human disturbance) to 1 (highest human disturbance). Our sampling stations represent a varied gradient of human modification from 0.04 in minimum to 0.28 in maximum, with a mean value of 0.12 (sd = 0.05). We quantified the level of acute direct human presence around each camera-trapping station by calculating the independent detections of human-related activities (e.g. livestock grazing, forest resource collection, and tourism) per 100 camera-trap days. The range of human presence recorded by our camera trapping survey was 0–46.81, with a mean value of 6.42 (sd = 8.61). We also chose a set of two environmental covariates known to impact spatiotemporal distribution of mammals (**Li et al., 2018**) and presumed to affect interspecific associations: (1) elevation, which plays a key role in shaping spatial distribution of many species in mountain forests (**He et al., 2019**; **Li et al., 2018**), and (2) percent of forest cover, which provides food resources, thermal cover and escape shelter for animals (**Long et al., 2005**). We derived percent of forest cover for each camera station based on the 250 m Moderate Resolution Imaging Spectroradiometer (MODIS) imagery (MOD44B Vegetation Continuous Fields (VCF) yearly product) of the study area for the period of 2021. For each camera trapping station, we derived human modification and percent of forest cover with a buffer radius of 500 m. This spatial scale should capture the environment that influences both resident animals with small home ranges and transient animals moving through the area. Prior to analysis, we log-transformed [log (x+0.1)] human presence data to account for its highly skewed distribution. We also standardized human modification and other environmental covariates by scaling to have a mean of zero and unit variance.

## Spatial co-occurrence analysis

We employed a context-dependent joint species distribution model (JSDM; **Tikhonov et al., 2017**) to characterize interspecific spatial associations of sympatric species. JSDMs are able to separate spatial associations between species into shared environmental preferences and residual correlations that cannot be explained by the environmental factors (**Pollock et al., 2014**). The context-dependent JSDM approach allows residual correlations to vary across the environment by incorporating species- and site-specific latent variables in the model (**Tikhonov et al., 2017**). We followed this approach and constructed a context-dependent model by utilizing a latent variable structure, where the factor loadings are modeled as a linear regression of covariates, allowing species associations to covary with human disturbance covariates. For our camera trap detections, we modeled the presence-absences of species j at camera station i as:

$$y_{ij} \, Bern\left(\Psi_{ij}\right)$$

with $\Psi_{ij} = \varphi^{-1}(\eta_{ij})$, where $\Psi_{ij}$ is the species-specific occurrence probability for each camera trapping station, and $\varphi^{-1}$ is the inverse of a probit link function. We modeled $\eta_{ij}$ as:

$$\eta_{ij} = \sum_{k=1}^{n_c} x_{ik}\beta_{jk} + \varepsilon_{ij}$$

where $n_c$ denotes the number of fixed covariates (i.e. elevation, percent of forest cover, human modification, and human presence) plus intercept, $\beta_{jk}$ denotes the effect of environmental covariate $k$ on species $j$, $x_{ik}$ denotes the measured covariates k=1… $n_c$ in the sampling unit $i$. The intercept of

the model is included by setting $x_{i1} = 1$ for all sampling units, so that the number of measured environmental covariates is $n_c - 1$. The species associations are modeled through the term $\varepsilon_{ij}$, which is defined by a latent factor model:

$$\varepsilon_{ij} = \sum_{h=1}^{n_f} z_{ih} \lambda_{jh} \left( x_{i\cdot}^* \right)$$

where $z_{ih}$ denotes the value of latent factor $h = 1 \ldots n_f$ at the sampling unit $i$, $\lambda_{jh} \left( x_{i\cdot}^* \right)$ denotes the response (factor loading) of species $j$ to latent factor $h$, given a vector of predictors $x_{i\cdot}^*$. The predictors $x_{i\cdot}^* = (x_{i1}^*, \ldots, x_{in_c^*}^*)$ on which the species associations are assumed to depend can be arbitrary, usually a subset of environmental predictors (*Tikhonov et al., 2017*). Here, we model the factor loadings of species as a function of the two types of human disturbances (i.e. human modification and human presence):

$$\lambda_{jh} \left( x_{i\cdot}^* \right) = \sum_{k=1}^{n_c^*} x_{ik}^* \lambda_{jhk}$$

where $n_c^*$ denotes the number of covariates assumed to impact residual correlations plus intercept. The intercept is included in the regression part by setting $x_{i1}^* = 1$ for all sampling units. In this study, assume $n_f = 3$, as our Deviance information criteria did not improve notably with the addition of more latent variables. We defined the covariance matrix of species factor loading as a function of human disturbances as $\varepsilon_{i\cdot} \sim N(0, \Omega(x_{i\cdot}^*))$, where $\Omega \left( x_{i\cdot}^* \right) = \Lambda \left( x_{i\cdot}^* \right) \Lambda \left( x_{i\cdot}^* \right)^T$, and $\Lambda \left( x_{i\cdot}^* \right)$ is the matrix of factor loadings, which depends on the human disturbances. We then scale this covariance matrix $\Omega$ to interspecific correlation matrices R by defining $R_{j1j2} = \Omega_{j1j2} / \sqrt{\Omega_{j1j1} \Omega_{j2j2}}$ for each pair of species, which represents disturbance-dependent associations between species that are not explained by fixed species-specific effects of environmental predictors. These resulted in values between –1 and 1, with negative values representing negative association between species, and positive values implying the opposite.

We estimated changes in species associations over continuous gradients of human modification (range: 0.04–0.28) and human presence gradients (range: 0.00–46.81). After fitting the model to data, we used the parameterized model to infer how species associations depend on human presence and human modification and generated predictions at minimum (lower), mean (moderate), and maximum (higher) conditions of the two variables separately.

We fitted the model based on a Bayesian approach using the greta R-package (*Golding, 2019*) as described by *Perrin et al., 2022*. We specified uninformative normally distributed priors for all parameters. We made inferences from 3000 samples on three chains after a burn-in of 2000 samples.

## Temporal co-occurrence analysis

For temporal co-occurrence analysis, we defined 'Low' and 'High' categories of human presence and human modification. We ranked camera trapping stations based on human presence and human modification separately, and filtered detections from the 25% most- ('High' category) and least- ('Low' category) disturbed camera trap stations and pooled detections within each category. We used kernel density distributions of animal diel activity and time between detections of species pairs (time-to-encounter) to compare temporal associations at lower and higher human disturbances. To examine if changes in species diel activity patterns were a mechanism behind anthropogenic impacts, we computed the nocturnal probability and time between consecutive detections of species pairs from lower- and higher-disturbance habitats. We transformed the detection time stamp to 'solar time' to eliminate the impacts of day-length variation in day length (*Nouvellet et al., 2012*). We retained only those species that had at least 30 independent detections in the low- and high-disturbance categories. For nocturnality analysis, we excluded detections within the hour around sunrise and sunset to avoid the effects of crepuscular activities on nocturnal probability. Thus, we defined nocturnal records as detections 1 hr after sunset up to 1 hr before sunrise. We conducted a binomial t-test to evaluate shifts in species nocturnality in the low- and high-disturbance categories.

To calculate time between detections of species pairs, we filtered camera stations to only those that detected at least two species for each disturbance category. We then calculated the time (in days) between successive detections across species pairs at each camera station. For analysis, we

log-transformed the time-to-encounter values to account for their markedly skewed distribution. We used 10,000 bootstrapped samples to evaluate 95% confidence interval (CI) shifts in time-to-encounter of species pairs from camera stations in the low- and high-disturbance categories. Results were considered to be significant if the key values fell outside of the 95% Confidence Interval.

## Acknowledgements

We thank the Tibetan Autonomous Region Forestry and Grassland Department for their logistic support and their authorities for permitting the study. We thank Minjing Pu, Kang Luo, Changzhe Pu, and other collaborators who helped collect data. We also thank Cheng Chen for his assistance with data management.

## Additional information

### Funding

| Funder | Grant reference number | Author |
|---|---|---|
| Second Tibetan Plateau Scientific Expedition and Research Program | STEP #2019 QZKK0501 | Xueyou Li |
| National Natural Science Foundation of China | #31601874 | Xueyou Li |
| West Light Foundation of the Chinese Academy of Sciences | | Xueyou Li |
| China Biodiversity Monitoring and Research Network | | Xueyou Li |
| Yunnan Provincial Youth Talent Support Program | YNWR-QNBJ-2020-127 | Xueyou Li |

The funders had no role in study design, data collection and interpretation, or the decision to submit the work for publication.

### Author contributions

Xueyou Li, Conceptualization, Data curation, Software, Formal analysis, Validation, Investigation, Visualization, Methodology, Writing – original draft, Writing – review and editing; William V Bleisch, Software, Validation, Visualization, Methodology, Writing – original draft, Writing – review and editing; Wenqiang Hu, Quan Li, Investigation; Hongjiao Wang, Zhongzheng Chen, Ru Bai, Data curation, Investigation; Xue-Long Jiang, Conceptualization, Supervision, Funding acquisition, Writing – original draft, Project administration, Writing – review and editing

### Author ORCIDs

Xueyou Li http://orcid.org/0000-0002-4705-6082
Quan Li http://orcid.org/0000-0001-7536-5475

Reviewer #1 (Public review): https://doi.org/10.7554/eLife.92457.3.sa1
Reviewer #2 (Public review): https://doi.org/10.7554/eLife.92457.3.sa2
Author response https://doi.org/10.7554/eLife.92457.3.sa3

## Additional files

### Supplementary files
• MDAR checklist

## Data availability

The authors declare that the data supporting the findings of this study are available within the article and its supplementary materials. Occupancy model data (independent detections of wildlife species and human activities histories and model covariates) are openly available in Science Data Bank: https://doi.org/10.57760/sciencedb.11804.

The following dataset was generated:

| Author(s) | Year | Dataset title | Dataset URL | Database and Identifier |
|---|---|---|---|---|
| Li X | 2023 | Camera trapping survey data at the Yarlung Zangbo Grand Canyon National Nature Reserve in the eastern Himalayas | https://doi.org/10.57760/sciencedb.11804 | Science Data Bank, 10.57760/sciencedb.11804 |

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
