## [Editor Report · eLife assessment]

In this study, camera trapping and species distribution models are used to show that human disturbance in mountain forests in the eastern Himalayas pushes medium-sized and large mammal species into narrower habitat space, thus increasing their co-occurrence. While the collected data provide a **useful** basis for further work, the study presents **incomplete** evidence to support the claim that increased co-occurrence may indicate positive interactions between species.

---

## [Referee Report · Reviewer #1 (Public review)]

Summary:

This study examines the spatial and temporal patterns of occurrence and the interspecific associations within a terrestrial mammalian community along human disturbance gradients. They conclude that human activity leads to a higher incidence of positive associations.

Strengths:

The theoretical framework of the study is brilliantly introduced. Solid data and sound methodology. This study is based on an extensive series of camera trap data. Good review of the literature on this topic.

Weaknesses:

The authors do not delve into the different types of association found in the study. A more ecological perspective explaining why certain species tend to exhibit negative associations and why others show the opposite pattern (and thus, can be used as indicator species) is missing. Also, the authors do not clearly distinguish between significant (true) non-random associations and random associations.

Anthropogenic pressures can shape species associations by increasing spatial and temporal co-occurrence, but above a certain threshold, the positive influence of human activity in terms of species associations could be reverted. This study can stimulate further work in this direction.

---

## [Referee Report · Reviewer #2 (Public review)]

Summary:

This study analyses camera trapping information on the occurrence of forest mammals along a gradient of human modification of the environment. The key hypotheses are that human disturbance squeezes wildlife into a smaller area or their activity into only part of the day, leading to increased co-occurrence under modification. The method used is joint species distribution modelling (JSDM).

Strengths:

The data source seems to be very nice, although since very little information is presented, this is hard to be sure of. Also, the JSDM approach is, in principle, a nice way of simultaneously analysing the data.

Weaknesses:

The manuscript suffers from a mismatch of hypotheses and methods at two different levels.

(1) At the lower level, we would need to better understand what the individual species do and "like" (their environmental niche).

(2) The hypothesis clearly asks for an analysis of the statistical interaction between human disturbance and co-occurrence. Yet, the study is not set up in a way to test this directly.

The hypotheses point towards presenting the spatial and the temporal niche, and how it changes, species for species, under human disturbance. To this, one could then add the layer of interspecific associations.

The change in activity and space use could be analysed by looking at the activity times and spatial distribution directly. If biotic interactions change along the disturbance gradient, then observed data are already the outcome of such changed interactions. We thus cannot use the data to infer them! But we can show, for each species, that the habitat preferences change along the disturbance gradient - or not, as the case may be.

The per-species models are simplistic: the predictors are only linear, and there are no statistical interactions. It is unclear how spatial autocorrelations of residuals were treated, although they form the basis for the association analysis. Why are times of day and day of the year not included as predictors IN INTERACTION with niche predictors and human disturbance, since they represent the temporal dimension on which niches are hypothesised to change?

The discussion has little to add to the results. The complexity of the challenge (understanding a community-level response after accounting for species-level responses) is not met, and instead substantial room is given to general statements of how important this line of research is. What is the advance in ecological understanding at the community level?

---

## [Author Response]

The following is the authors’ response to the original reviews.

**Reviewer #1 (Public Review):**
Summary:This study examines the spatial and temporal patterns of occurrence and the interspecific associations within a terrestrial mammalian community along human disturbance gradients. They conclude that human activity leads to a higher incidence of positive associations.Strengths:The theoretical framework of the study is brilliantly introduced. Solid data and sound methodology. This study is based on an extensive series of camera trap data. Good review of the literature on this topic.Weaknesses:The authors use the terms associations and interactions interchangeably.

This is not the case. In fact, we state specifically that "... interspecific associations should not be directly interpreted as a signal of biotic interactions between pairs of species…" However, co-occurrence can be an important predictor of likely interactions, such as competition and predation. We stand by our original text.

It is not clear what the authors mean by "associations". A brief clarification would be helpful.

Our specific definition of what is meant here by spatial association can be found in the Methods section. To clarify, the calculation of the index of associations is based on the covariance for the two species of the residuals (epsilon) after consideration of all species-specific response to known environmental covariates. These covariances are modelled to allow them to vary with the level of human disturbance, measured as human presence and human modification. After normalization, the final index of association is a correlation value that varies between -1 (complete disassociation) and +1 (complete positive association).

Also, the authors do not delve into the different types of association found in the study. A more ecological perspective explaining why certain species tend to exhibit negative associations and why others show the opposite pattern (and thus, can be used as indicator species) is missing.

Suggesting the ecological underpinnings of the associations observed here would mainly be speculation at this point, but the associations demonstrated in this analysis do suggest promising areas for the more detailed research suggested.

Also, the authors do not distinguish between significant (true) non-random associations and random associations. In my opinion, associations are those in which two species co-occur more or less than expected by chance. This is not well addressed in the present version of the manuscript.

Results were considered to be non-random if correlation coefficients (for spatial association) or overlap (for temporal association) fell outside of 95% Confidence Intervals. This is now stated clearly in the Methods section. In Figure 3—figure supplement 1-3 and Figure 4—figure supplement 1-3, p<0.01 levels are also presented.

The obtained results support the conclusions of the study.Anthropogenic pressures can shape species associations by increasing spatial and temporal co-occurrence, but above a certain threshold, the positive influence of human activity in terms of species associations could be reverted. This study can stimulate further work in this direction.
**Reviewer #2 (Public Review):**
Summary:This study analyses camera trapping information on the occurrence of forest mammals along a gradient of human modification of the environment. The key hypotheses are that human disturbance squeezes wildlife into a smaller area or their activity into only part of the day, leading to increased co-occurrence under modification. The method used is joint species distribution modelling (JSDM).Strengths:The data source seems to be very nice, although since very little information is presented, this is hard to be sure of. Also, the JSDM approach is, in principle, a nice way of simultaneously analysing the data.Weaknesses:The manuscript suffers from a mismatch of hypotheses and methods at two different levels.(1) At the lower level, we first need to understand what the individual species do and "like" (their environmental niche). That information is not presented, and the methods suggest that the representation of each species in the JSDM is likely to be extremely poor.

The response of each species to the environmental covariates provides a window into their environmental niche, encapsulated in the beta coefficients for each environmental covariate. This information is presented in Figure 2.

(2) The hypothesis clearly asks for an analysis of the statistical interaction between human disturbance and co-occurrence. Yet, the model is not set up this way, and the authors thus do a lot of indirect exploration, rather than direct hypothesis testing.

Our JSDM model is set up specifically to examine the effect of human disturbance on co-occurrence, after controlling for shared responses to environmental variables. It directly tests the first hypothesis, since, if increase in indices of human disturbance had not tended to increase the measured spatial correlations between species as detected by the model, we would have rejected our stated hypothesis that human modification of habitats results in increased positive spatial associations between species.

Even when the focus is not the individual species, but rather their association, we need to formulate what the expectation is. The hypotheses point towards presenting the spatial and the temporal niche, and how it changes, species for species, under human disturbance. To this, one can then add the layer of interspecific associations.

Examining each species one by one and how each one responds to human disturbance would miss the effects of any meaningful interactions between species. The analysis presented provides a means to highlight associations that would have been overlooked. Future research could go on to analyze the strongest associations in the community and the strongest effects of human disturbance so as to uncover the underlying interactions that give rise to them and the mechanisms of human impact. We believe that this will prove to be a much more productive approach than trying to tackle this problem species by species and pair by pair.

The change in activity and space use can be analysed much simpler, by looking at the activity times and spatial distribution directly. It remains unclear what the contribution of the JSDM is, unless it is able to represent this activity and spatial information, and put it in a testable interaction with human disturbance.The topic is actually rather complicated. If biotic interactions change along the disturbance gradient, then observed data are already the outcome of such changed interactions. We thus cannot use the data to infer them! But we can show, for each species, that the habitat preferences change along the disturbance gradient - or not, as the case may be.Then, in the next step, one would have to formulate specific hypotheses about which species are likely to change their associations more, and which less (based e.g. on predator-prey or competitive interactions). The data and analyses presented do not answer any of these issues.

We suggest that the so-called “simpler” approach described above is anything but simple, and this is precisely what the Joint Species Distribution Model improves upon. As pointed out in the Introduction, simply examining spatial overlap is not enough to detect a signal of meaningful biotic interaction, since overlap could be the result of similar responses to environmental variables. With the JSDM approach, this would not be considered a positive association and would then not imply the possible existence of meaningful interaction.

Another more substantial point is that, according to my understanding of the methods, the per-species models are very inappropriate: the predictors are only linear, and there are no statistical interactions (L374). There is no conceivable species in the world whose niche would be described by such an oversimplified model.

While interaction terms can be included in the JSDM, this would considerably increase the complexity of the models. In previous work, we have found no strong evidence for the importance of interaction terms and they do not improve the performance of the models.

We have no idea of even the most basic characteristics of the per-species models: prevalences, coefficient estimates, D2 of the model, and analysis of the temporal and spatial autocorrelation of the residuals, although they form the basis for the association analysis!

The coefficient estimates for response to environmental variables used in the JSDM are provided in Figure 2 and Figure 2—source data 1.

Why are times of day and day of the year not included as predictors IN INTERACTION with niche predictors and human disturbance, since they represent the temporal dimension on which niches are hypothesised to change?Also, all correlations among species should be shown for the raw data and for the model residuals: how much does that actually change and can thus be explained by the niche models?The discussion has little to add to the results. The complexity of the challenge (understanding a community-level response after accounting for species-level responses) is not met, and instead substantial room is given to general statements of how important this line of research is. I failed to see any advance in ecological understanding at the community level.

We agree that the community-level response to human disturbance is a complex topic, and we believe it is also a very important one. This research and its support of the spatial compression hypothesis, while not providing definitive answers to detailed mechanisms, opens up new lines of inquiry that makes it an important advance. For example, the strong effects of human disturbance on certain associations that were detected here could now be examined with the kind of detailed species by species and pair by pair analysis that this reviewer appears to demand.

**Reviewer #1 (Recommendations For The Authors):**
L27 indicates instead of "idicates".

We thank the reviewer for catching that error.

L64 I would refer to potential interactions or just associations. It is always hard to provide evidence for the existence of true interactions.

We have revised to “potential interactions” to qualify this statement.

L69 Suggestion: distort instead of upset.

We thank the reviewer for catching that error.

L70-71 Here, authors use the term associations. Please, be consistent with the terminology throughout the manuscript.

We thank the reviewer for raising this important point. The term “co-occurrence” appears to be used inconsistently in the literature, so we have tried to refer to it only when referencing the work of us. For us, co-occurrence means “spatial overlap” without qualification as to whether it is caused by interaction or simply by similar responses to environmental factors (see Blanchet et al. 2020, Argument 1). In our view, interactions refer to biotic effects like predation, competition, commensalism, etc., while associations are the statistical footprint of these processes. In keeping with this understanding, in Line 73, we changed "association" to the stronger word "interaction," but in Line 76, we keep the words "spatiotemporal association", which is presumed to be the result of those interactions. In Line 91, we have changed “interactions” to “associations,” as we do not believe interactions were demonstrated in that study.

L76 "Species associations are not necessarily fixed as positive or negative..." This sentence is misleading. I would say that species associations can vary across time and space, for instance along an environmental gradient.

We thank the reviewer for pointing out the potential for confusion. In Line 79, we have changed as suggested.

L78 "Associations between free-ranging species are especially context-dependent" Loose sentence. Please, explain a bit further.

We have changed the sentence to be more specific; ”Interactions are known to be context-dependent; for example, gradients in stress are associated with variation in the outcomes of pairwise species interactions.”

L83-85 This would be a good place to introduce the 'stress gradient' hypothesis, which has also been applied to faunal communities in a few studies. According to this hypothesis, the incidence of positive associations should increase as environmental conditions harden.

In our review of the literature, we find that the stress gradient hypothesis is somewhat controversial and does not receive strong support in vertebrates. We have added the phrase “…the controversial stress-gradient hypothesis predicts that positive associations should increase as environmental conditions become more severe…”

L86-88 Well, overall, the number of studies examining spatiotemporal associations in vertebrates is relatively small. That is, bird associations have not received much more attention than those of mammals. I find this introductory/appealing paragraph a bit rough. I think the authors can do better and find a better justification for their work.

We thank the reviewer for the comments. We have rewritten the paragraph extensively to make it clearer and to provide a stronger justification for the study.

L106 "[...] resulting in increased positive spatial associations between species" I'd say that habitat shrinking would increase the level of species clustering or co-occurrence, but in my opinion, not necessarily the incidence of positive associations. It is not clear to me if the authors use positive associations as a term analogous to co-occurrence.

We thank the reviewer for raising this very important distinction. Habitat shrinking would increase levels of species co-occurrence, but this is not particularly interested. We wanted to test whether there were effects on species interactions, as revealed by associations. We find that the terms association and co-occurrence are used somewhat loosely in the literature and so have made some new effort to clarify and systematize this in the manuscript. For example, there appear to be a differences in the way “co-occurrence” is used in Boron 2023 and in Blanchet 2020. We do not use the term "positive spatial association" as analogous to "spatial co-occurrence.". Spatial co-occurrence, which for us has the meaning of spatial overlap, could simply be the result of similar reactions to environmental co-variates, not reflecting any biotic interaction. Joint Species Distribution Models enable the partitioning of spatial overlap and segregation into that which can be explained by responses to known environmental factors, and that which cannot be explained and thus might be the result of biotic interactions. It is only the latter that we are calling spatial association, which can be positive or negative. These associations may be the statistical footprint of biotic interactions.

Results:Difference between random and non-random association patterns. It is not clear to me if the reported associations are significant or not. The authors only report the sign of the association (either positive or negative) but do not clarify if these associations indicate that two species coexist more or less than expected by chance. In my opinion, that is the difference between true ecological associations (e.g., via facilitation or competition effects) and random co-existence patterns. This is paramount and should be addressed in a new version of the manuscript.

This information is provided in Figure 3—figure supplement 1,2,3 and Figure 4—figure supplement 1,2,3. This is referenced in the text as follows, “… correlation coefficients for 18 species pairs were positive and had a 95 % CI that did not overlap zero, and the number increased to 65 in moderate modifications but dropped to 29 at higher modifications" and so on. This criterion for significance (ie., greater than expected by chance) is now stated at the end of the Materials and methods. In Figure 3—figure supplement 1,2,3 and Figure 4—figure supplement 1,2,3, those correlations that were significant at p<0.01 are also shown.

I am also missing a more ecological explanation for the observed findings. For instance, the top-ranked species in terms of negative associations is the red fox, whereas the muntjac seems to be the species whose presence can be used as an indicator for that of other species. What are the mechanisms underlying these patterns? Do red foxes compete for food with other species? Do the species that show positive associations (red goral, muntjac) have traits or a diet that are more different from those of other species? More discussion on these aspects (role of traits and the trophic niche) would be necessary to better understand the obtained results.

The purpose of this paper was to test the compression hypotheses, and we have tried to keep that as the focus. However, the analysis does open up interesting lines of inquiry for future research to decipher the details of the interactions between species and the mechanisms by which human disturbance facilitates or disrupts these interactions. The reviewer raises some interesting possibilities, but at this point, any discussion along these lines would be largely speculation and could lengthen the paper without great benefit.

**Reviewer #2 (Recommendations For The Authors):**
The manuscript should be accompanied by all data and code of analysis.All data and RScripts have been made available in Science Data Bank: https://doi.org/10.57760/sciencedb.11804.The sentence "not much is known" is weak: it suggests the authors did not bother to quantify what IS known, and simply waved any previous knowledge aside. Surely we have some ideas about who preys on whom, and which species have overlapping resource requirements (e.g., due to jaw width). For those, we would expect a particularly strong signal, if the association is indeed indicative of interactions.

We believe that the reviewer is referring to the statement in Line 90-92 about the lack of understanding of the resilience of terrestrial mammal associations to human disturbance. We have added a reference to one very recent publication that addresses the issue (Boron et al., 2023), but otherwise we stand by our statement. We have, however, added a qualifier to make it clear that we did indeed look for previous knowledge; "However, a review of the literature indicates that ...."

Figures:Fig. 1. This reviewer considers that this is too trivial and should be deleted.

This is a graphical statement of the hypotheses and may be helpful to some readers.

Fig. 2. Using points with error bars hides any potential information.

Done as suggested.

That only 4 predictors are presented is unacceptably oversimplified.

Only 4 predictors are included because, in previous work, we found that adding additional predictors or interactions did little to improve the model’s performance (Li et al. 2018, 2021 and 2022) and could lead to over-fitting.

Fig. 5. and 6. aggregate extremely strongly over species; it remains unclear which species contribute to the signal, and I guess most do not.

The number of detection events presented in Table 1 should help to clarify the relative contribution of each species to the data presented in Figures 5 and 6.

This reviewer considers that the introduction 'oversells' the paper.L55: can you give any such "unique ecological information"L60: Lyons et al. (Kathleen is the first name) has been challenged by Telford et al. (2016 Nature) as methodologically flawed.

The first name has been deleted. The methodological flaw has to do with interpretation of the fossil record and choice of samples, not with the need to partition shared environmental preferences and interactions.

L61 contradicts line 64: Blanchet et al. (2022, specifying some arguments from Dormann et al. 2018 GEB) correctly point out that logically one cannot infer the existence or strength from co-occurrence data. It is thus wrong to then claim (citing Boron et al.) that such data "convey key information about interactions". The latter statement is incorrect. A tree and a beetle can have extremely high association and nothing to do with each other. Association does not mean anything in itself. When two species are spatially and temporally non-overlapping, they can exhibit perfect "anti-association", yet, by the authors' own definition, cannot interact.

We believe that the reviewer’s concerns arise from a misunderstanding of how we use the term association. In our usage, an association is not the same as co-occurrence or overlap, which may simply be the result of shared responses to environmental variables. The co-occurring tree and beetle would not be found to have any association in our analysis, only shared environmental sensitivities. In contrast, associations can be the statistical footprint of interactions, and would be overlaid onto any overlap due to similar responses to the environment. In the case of negative associations, such as might be the result of competitive exclusion or avoidance of predators, the two species would share environmental responses but show lower than expected spatial overlap. Even though they might be only rarely found in the same vicinity, they would indeed be interacting when they were together.

Joint Species Distribution Models "allow the partitioning of the observed correlation into that which can be explained by species responses to environmental factors... and that which remains unexplained after controlling for environmental effects and which may reflect biotic interactions." (Garcia Navas et al. 2021). It is the latter that we are calling “associations.”

L63: Gilbert reference: Good to have a reference for this statement.

This point is important, but the reviewer’s comments below have made it clear that it is even more important to point out that strong interactions should be expected to lead to significant associations. We have added a statement to clarify this.

L70-72: Incorrect, interactions play a role, not associations (which are merely statistical).

In this, we agree, and we have revised the statement to refer to interactions, not associations. In our view, an interaction is a biological phenomenon, while an association is the resulting statistical signal that we can detect.

L75: Associations tell us nothing, only interactions do. Since these can not be reliably inferred, this statement and this claim are wrong.

We thank the reviewer for raising this point, but we beg to disagree. Strong interactions should be expected to lead to significant associations that can be detected in the data. Associations, which can be measured reliably, are the evidence of potential interactions, and hence associations can tell us a great deal. We have added a note to this effect after the Gilbert reference above to clarify this point.

However, we do accept that associations must be interpreted with caution. As Blanchet et al. 2020 explain, " …the co-occurrence signals (e.g. a signiﬁcant positive or negative correlation value) estimated from these models could originate from any abiotic factors that impact species differently. Therefore, this correlation cannot be systematically interpreted as a signal of biotic interactions, as it could instead express potential non-measured environmental drivers (or combinations of them) that inﬂuence species distribution and co-distribution.” Or alternatively an association could be the result of interaction with a 3rd species.

L87: Regarding your claim, how would you know you DO understand? For that, you need to formulate an expectation before looking at the data and then show you cannot show what you actually measure. (Jaynes called this the "mind-projection fallacy".)

We are not sure if the reviewer is criticizing our paper or the entire field of community ecology. Perhaps it is the statement that “….resilience of interspecific spatiotemporal associations of terrestrial mammals to human activity remains poorly understood….” Since we are confident that the reviewer believes that mammals do interact, we guess that it is the term “association” that is questioned. We have revised this to “…the impacts of human activity on interspecific interactions of terrestrial mammals remains poorly understood…”

In this particular case, we did formulate an expectation before looking at the data, in the form of the two formal hypotheses that are clearly stated in the Introduction and illustrated in Figure 1. If the hypotheses had not been supported, then we would have accepted that we do not understand. But as the data are consistent with the hypotheses, we submit that we do understand a bit more now.